# Role of the Alteration in Calcium Homeostasis in Cell Death Induced by *Clostridioides difficile* Toxin A and Toxin B

**DOI:** 10.3390/biology12081117

**Published:** 2023-08-10

**Authors:** Katia Fettucciari, Fabrizio Dini, Pierfrancesco Marconi, Gabrio Bassotti

**Affiliations:** 1Biosciences & Medical Embryology Section, Department of Medicine and Surgery, University of Perugia, 06129 Perugia, Italy; pierfrancesco.marconi@outlook.it; 2School of Biosciences and Veterinary Medicine, University of Camerino, 62024 Matelica, Italy; fabrizio.dini@unicam.it; 3Gastroenterology, Hepatology & Digestive Endoscopy Section, Department of Medicine and Surgery, University of Perugia, 06129 Perugia, Italy; gabassot@gmail.com; 4Gastroenterology & Hepatology Unit, Santa Maria Della Misericordia Hospital, 06129 Perugia, Italy

**Keywords:** *Clostridioides difficile*, Clostridioides difficile toxin A, *Clostridioides difficile* toxin B, pore-forming toxins, cell death, calcium, cytotoxicity, apoptosis, necrosis, pyroptosis

## Abstract

**Simple Summary:**

*Clostridioides difficile* (*C. difficile*) infections represent a major health risk and a high cost to health-care systems. *C. difficile* causes pathogenic activity in the colon through the production of two toxins, A and B, which provoke cell death. Several studies conducted on the mechanisms by which these two toxins induce various types of cell death in the intestinal mucosa cells and in the immune system cells failed to identify a common basic event. Since calcium is an important regulator of cellular physiology, here we propose a new point of view in which the alteration in physiological intracellular calcium levels by the two toxins may be the key common event responsible for the activation of cell death. Further exploration of this issue with in vivo studies could be important for possible pharmacological interventions aimed at antagonizing the Ca^2+^ homeostasis alterations that could tone down the *C. difficile* pathogenicity until specific antibiotic therapy has brought the infection under control.

**Abstract:**

*Clostridioides difficile* (*C. difficile*), responsible for 15–25% of gastrointestinal infections, causes health problems mainly due to the toxic activity of toxins A and B (Tcds). These are responsible for its clinical manifestations, including diarrhea, pseudomembranous colitis, toxic megacolon and death, with a mortality of 5–30% in primary infection, that increase following relapses. Studies on Tcd-induced cell death have highlighted a key role of caspases, calpains, and cathepsins, with involvement of mitochondria and reactive oxygen species (ROS) in a complex signaling pathway network. The complex response in the execution of various types of cell death (apoptosis, necrosis, pyroptosis and pyknosis) depends on the amount of Tcd, cell types, and Tcd receptors involved, and could have as initial/precocious event the alterations in calcium homeostasis. The entities, peculiarities and cell types involved in these alterations will decide the signaling pathways activated and cell death type. Calcium homeostasis alterations can be caused by calcium influx through calcium channel activation, transient intracellular calcium oscillations, and leakage of calcium from intracellular stores. These increases in cytoplasmic calcium have important effects on all calcium-regulated molecules, which may play a direct role in several cell death types and/or activate other cell death effectors, such as caspases, calpains, ROS and proapoptotic Bcl-2 family members. Furthermore, some support for the possible role of the calcium homeostasis alteration in Tcd-induced cell death originates from the similarity with cytotoxic effects that cause pore-forming toxins, based mainly on calcium influx through plasma membrane pores.

## 1. Introduction

*Clostridioides difficile* (*C. difficile*) [1,2] is a Gram-positive spore-forming bacterium responsible for 15–25% of all opportunistic gastrointestinal infections [3,4,5]. The clinical manifestations of *C. difficile* infection (CDI) range from mild transient diarrhea to pseudomembranous colitis and toxic megacolon [6,7]. The pathogenic activity of *C. difficile* is mainly due to two exotoxins, toxin A (TcdA) and toxin B (TcdB), with TcdB being about 1000 times more potent than TcdA on cytotoxic activity. A minor pathogenic contribution is also due to the binary toxin [8,9,10,11,12].

At present, CDIs have become a serious health problem because they are not only frequent in subjects undergoing antibiotic therapy or in those hospitalized/institutionalized but also because the incidence is increasing in the community, in both elderly and young subjects, in those with severe illnesses e.g., inflammatory bowel disease (IBD) and also as a consequence of increased feeding tube use and other surgical or therapeutic practices [13,14,15,16,17]. Furthermore, CDI spread is aggravated by the continuous emergence of new hypervirulent *C. difficile* strains [18,19], the progressive diffusion of *C. difficile* in the anthropized environment, and the increasing ability to colonize humans [20].

*C. difficile* is transmitted person-to-person via the fecal–oral route by spores, and hospitals and community health-care settings may become an important source of infection [6,7,21]. Viable *C. difficile* spores can be found almost ubiquitously in hospitals and nursing home environment, health-care workers’ hands, and several medical devices [6,7,20,21]. Operative strategies to lower the CDI overall burden in hospitals and nursing homes together with a decrease in antibiotic therapy include patient insulation, hygiene/washing hands, use of disposable gloves and gowns, deep cleansing of room/surfaces, and full room decontamination after discharge [22,23]. Nurses together with other health-care workers are involved in several important aspects of CDI patient care and are crucial active participants in the correct employment of infection control practices aimed at lowering the CDI burden in health-care settings. Recently, a substantial lack of information on CDI in a population of Italian nurses regarding CDI clinical guidelines, handwashing practices, and environmental and contamination routines has been documented [23]. Therefore, education of health-care professionals including nurses is of paramount importance to reduce transmission of *C. difficile* and hospital outbreaks [23].

So far, attempts to develop non-antibiotic therapies against CDI based on antibodies [24,25], vaccines [26,27], fecal microbiota transplantation [28,29,30], polymeric binders [31], inhibitors of spore germination [32,33], non-toxigenic *C. difficile* colonization [34], *C. difficile* bacteriophages (phages) [35,36] and inositol hexakisphosphate (IP6) modified to inactivate TcdB before its entry into the cell [37] have given limited results or suffer from methodological limitations [38,39,40,41].

Consequently, it is necessary to gain further knowledge of the basic mechanisms underlying the pathogenic activity of TcdA and TcdB to find in the network of activated signaling pathways molecular targets to antagonize in order to inhibit the pathogenic activity of *C. difficile* toxins (Tcds).

As we discuss below, within the pathogenic mechanism of CDI, there is an early phase of about 40 min duration, characterized by the alteration in calcium (Ca^2+^) homeostasis followed by the inhibition of Rho-GTPases. This first phase, still poorly understood, is perhaps decisive in the induction of cell death and for the type of cell death and could be of crucial importance to understand the manner of antagonizing the key pathogenic events of *C. difficile*.

In this review, we discuss in depth several significant aspects that indicate, albeit indirectly, that in the mechanism of CDI pathogenicity, there is an early phase characterized by the alteration in Ca^2+^ homeostasis, followed by the inhibition of Rho-GTPases, possibly contributing per se to further Ca^2+^ homeostasis alterations, and highlighted that is the kinetic of Ca^2+^ homeostasis alteration that leads to different cytotoxic effects or to different types of cell death (e.g., apoptosis, necrosis, pyknosis or pyroptosis). Furthermore, we compare the effects of pore-forming toxins (PFTs) and Tcds, because an additional indirect enforcement of the possible role of the alteration in Ca^2+^ homeostasis in cell death induced by Tcds comes from the similarity with the effects that PFTs cause during induction of cell death, based mainly on the direct Ca^2+^ influx by plasma membrane pore formation.

The comparison between PFTs and Tcds takes into account the fact that PFTs form a stable pore in the membrane. This pore is the main responsible for the influx of Ca^2+^ with all the consequences due to the profound alteration in intracytoplasmic Ca^2+^ homeostasis, while the pathogenic activity of Tcds until now has been characterized only as the activation of Rho-GTPase-dependent and Rho-GTPase-independent mechanisms after internalization of the Tcd (see below) [8,9,10,11,12].

Overall, for the first time, the possibility that Ca^2+^ homeostasis alteration could be a key event in the mechanism of pathogenicity of *C. difficile* is discussed and highlighted. Knowledge of this aspect could thus be of crucial importance to understand an alternative manner to antagonize the pathogenicity of *C. difficile*.

## 2. *C. difficile*

### 2.1. General Characteristics of CDI

Once the spores of *C. difficile* have reached the colon, if the right conditions are present, among which the most relevant is the alteration in the microbiota, they enter the vegetative phase with the dissolution of the spore wall and the production of TcdA and TcdB and, in some strains, also of binary toxin [6,7,42,43]. TcdA and TcdB induce death mainly by apoptosis and/or necrosis in all the colonic cell types and of inflammatory immune cells involved [8,9,10,11,12]. Thus, in the context of a strong inflammatory response, *C. difficile* penetrates more deeply into the mucosa [8,9,10].

Although other molecular components of *C. difficile* may contribute to pathogenicity [44,45], the key role is played by the ability of its Tcds to cause cell death [8,9,10,11,12,46]. Studies on the mechanisms of Tcd-induced cell death yielded interesting results, leading to the identification of the main signaling pathways responsible for different types of cell death [8,9,10,11,12,46,47].

However, little emphasis has been placed on the possible key role of alteration in homeostasis of intracellular Ca^2+^ as an early key event in the process of cell death.

### 2.2. Molecular Structure of Tcds

TcdA and TcdB belong to the large clostridial toxins family, which also includes the toxins produced by *C. perfrigens* and *C. novi* [48]. TcdB is primarily responsible for the clinical manifestations of CDI [8,9,10,11,49]. Tcds are multifunctional proteins due to the complex molecular organization of the domains of which they are constituted and the peculiar conformational responses to changes in pH and after interaction with ligands [8,9,10,11]. This multifunctionality is also due to the contribution of intrinsically disordered regions of the Tcd [50]. TcdA consists of 2710 amino acids, while TcdB consists of 2366 amino acids. They have 66% sequence similarity and 44% sequence identity [8,9,10,11].

Tcds present both a analogous glucosyltransferase enzymatic activity and a similar structural composition with multi-modular domains, referred as the ABCD model (module A: biological activity; module B: binding; module C: cutting; module D: delivery) [8,9,10,11]. Module A corresponds to the N-terminal glucosyltransferase domain (GTD) that alters the cytoskeleton by monoglucosylation of small GTPase of the Rho family. Module B corresponds to the C-terminal-located receptor-binding region that includes the combined repetitive oligopeptide (CROP) domain, a relevant part of the receptor-binding domain. Module C corresponds to the autoprocessing domain (APD) or cysteine protease domain (CPD), responsible for autocatalytic cleavage. Module D corresponds to the translocation/pore-forming domain or delivery domain responsible for the translocation of the catalytic domain into the cytoplasm from the endocytic vacuole. This domain is also involved in the binding of Tcds to receptors on target cells [8,9,10,11].

### 2.3. Tcd Receptors

The receptors described for TcdA are two proteins expressed in the cell plasma membrane [8,9,51]. The first TcdA receptor is the sucrase isomaltase, a glycoprotein expressed in the small intestine brush border [52], but is not expressed in several cells and tissues that are susceptible to TcdA, such as the human colonic epithelium [52]. The second is glycoprotein 96 (gp96), a heat shock protein family member [53]. Cells that not expressed the gp96 receptor are only partly resistant to TcdA intoxication, implying that TcdA binds other receptors [53].

The receptors described for TcdB number three, and are proteins expressed in cell cytoplasmic membrane [8,9,51]. The first is chondroitin sulphate proteoglycan 4 (CSPG4) [54]; the binding region is localized at the N-terminal of CROPs [54,55] but other evidence has revealed that the function of binding to the receptor is not restricted to this domain of the Tcd. In fact, it seems that the CROPs participate in but are not necessary for the binding to host cells [55,56]. CSPG4 is strongly expressed in the intestinal subepithelial myofibroblasts, but not on the surface of the epithelium [55,57]. The second TcdB receptor is NECTIN3, also named poliovirus receptor-like protein (PVRL3), and the third TcdB receptor is frizzled proteins 1, 2 and 7 (FZD1,2,7) [51,55,56,58]. These are expressed on the surface epithelium of the human colon, and so are both colonic epithelial receptors for TcdB [55,56]. Either PVRL3 or FZDs bind TcdB beyond the CROPs [55,56]. It is still unknown if FZDs and PVRL3 bind TcdB in diverse regions or contend for the same region. In contrast, it has been reported that binding to CSPG4 and to FZD occurs independently and in an additive manner [58].

### 2.4. Cellular Uptake of Tcds and Tcd-Induced Cellular Effects

TcdA and TcdB, after binding to the cell plasma membrane receptors for Tcd, prompt its uptake by different pathways. Uptake of TcdB is mediated by clathrin, while that of TcdA is mediated by PACSIN2/syndapin-II [8,59,60]. Following Tcd uptake in the endocytic vacuole, the increase in pH acidity elicits conformational modifications of Tcd that promote the insertion of catalytic domain of Tcds externally to the membrane for its cleavage favored by the cell’s IP6 and translocation into the cytoplasm. Then, Tcds glucosylate Rho-GTPase at the catalytic site, inhibiting their functions [8,9,10,11,60]. Glucosylation of Rho-GTPase induces several important biological effects, such as [8,9,10,11,12,46,60,61]: (1) early cytoskeletal alteration, dismantlement of focal adhesion and tight-junction disruption, which in vitro result in cell rounding (cytopathic effect); (2) cell cycle arrest, associated with reduced expression of the cyclin D1/Cdk4 or cyclin B1/Cdk1 complex, which are required for progression by the G1 or G2 phase, respectively, resulting in G1/S or G2/M arrest; and (3) cell death (cytotoxic effect), which occurs after the cycle arrest, mainly by apoptosis and necrosis, though it may also occur by pyknosis or pyroptosis.

Of interest, TcdA and TcdB may also induce cell death (cytotoxic effect) independently of autoprocessing or glucosyltransferase activities of the Tcds, which occurs mainly by necrosis, but can also occur by apoptosis, pyknosis, or pyroptosis (see below) [8,9,62,63,64,65].

Tcds are able to cause death in several types of cells: colonocytes [57], enterocytes [66] myocytes [67], enteric neurons [68], enteric glial cells (EGCs) [46], immune cells [69,70,71], nervous cells [72,73], lung fibroblasts [74], hepatic cells [75] and cardiac cells [76]. Tcds also trigger the release of a variety of inflammatory mediators, in particular from epithelial cells, immune cells and EGCs [8,9,10,11,77,78,79].

Notably, Fettucciari et al. showed that EGCs that survive the TcdB cytotoxic effects go into a senescence state as a survival response to a stressor stimulus prompted by TcdB [46,80].

Although glucosylation-independent necrosis has been observed in vitro, the relevance of this mechanism during CDI has remained an outstanding question. Recently, Peritore-Galve et al. showed that in a mouse model of CDI, epithelial damage occurs through a glucosylation-independent manner not mediated by immune cell influx. In particular, the authors showed that the glucosylation activity of Tcds causes severe edema and inflammation and recruitment of immune cells. Glucosylation activity of both Tcd was necessary to produce weight loss and severe watery diarrhea, while it was not necessary to cause epithelial damage and pseudomembrane formation. Moreover, glucosylation-independent epithelial damage did not require/elicit recruitment of immune cells [81].

## 3. Mechanisms of Tcd-Induced Cytotoxicity

Overall, the cytotoxic activity of Tcds, which are multifunctional proteins with intrinsically disordered regions [50], is due to several properties: lipid binding, cysteine-type peptidase activity, IP6 binding, metal ion binding, peptidase activity, transferase activity mainly based on ability to transfer glucosyl groups, UDP-glucosyltransferase activity, and finally the ability to alter early the Ca^2+^ homeostasis [8,9,10,11,37,82].

The most evident manifestations of the inactivation by glucosylation of the Rho-GTPase members Rac1, Cdc42 and Rho by Tcds are the cytopathic effects (cell rounding) due to derangements of the actin cytoskeleton with cell cycle arrest and cytotoxic effects that cause cell death [8,9,10,11,12,46,61].

The cytotoxic activity induced by TcdB is manifested in all cell types, while this is not the case for TcdA, probably due to the greater cytotoxic potency of TcdB and the different types of membrane receptors targeted by the two Tcds [8,9,10,46]. In fact, receptors for TcdB are predominantly CSPG4, PVRL3, and FZDs [8,9,51,54,55,56,57], while those for TcdA are predominantly sucrase isomaltase and gp96 [8,9,51,52,53]. Another significant distinctiveness of TcdB is that it can bind to the membrane receptor with amino acid sequences that extend beyond the CROP sequences of receptor binding domains [55,56].

Furthermore, the two Tcds show differences in effects at low and high concentrations. In fact, when TcdB is present at low concentrations (picomolar concentrations) and binds to membrane receptors CSPG4 and FZD, it induces mainly apoptosis in a glucosylation-dependent way, while when TcdB is present at higher concentrations (the nanomolar concentrations) and binds to CPSG4, FZD and PVL3 receptors, it induces mainly necrosis in a glucosylation- and autoprocessing-independent way [51,54,55,56,57,63].

Studies on the mechanisms by which TcdA [66,83,84,85,86,87] or TcdB [46,47,72,88,89,90,91,92] induces apoptosis have shown a central role played by caspases. In fact, it has been demonstrated in several different cell types that the executioner caspase-3 plays a key role in TcdA- and TcdB-induced apoptosis [8,9,10,11,46,47,60] but activation of the executioners caspase-6 and caspase-7 was also involved [46,47,60,66,85].

Effector caspase activation by Tcds is mediated through an extrinsic death pathway dependent on receptors that encompasses caspase-8, or through an intrinsic death pathway dependent on mitochondria that include caspase-9 [9]. By increasing the levels of proapoptotic B-cell lymphoma 2 (Bcl-2) members and/or decreasing the levels of antiapoptotic Bcl-2 members, TcdA and TcdB cause mitochondrial outer membrane permeabilization (MOMP) with consequent release of cytochrome C and caspase-9 activation by a mitochondria-dependent pathway [9,11]. TcdA and TcdB also activate caspase-8, which triggers apoptosis by a receptor-dependent or by a mitochondria-dependent death pathway [46,47,66,85,86,87,93,94]. Even though Tcds cause caspase activation, other effector mechanisms are also involved. In fact, by means of caspase inhibitors and glucosyltransferase-deficient mutants, it has been demonstrated that apoptotic mechanisms, both caspase-dependent and caspase-independent, are activated for both TcdA and TcdB [47,63,66,86,87,89,91,94]. In many apoptotic pathways, MOMP is controlled by alterations in the expression/activation of pro- and antiapoptotic Bcl-2 family members, but it can also trigger apoptosis in a caspase-independent way. In fact, MOMP is caused by cleavage and activation of Bid (a proapoptotic member of the Bcl-2 family), cleaved by caspase-8, but also by non-caspase proteases (e.g., calpains and cathepsins) [9,47,86,95,96,97,98,99,100,101]. Interestingly, TcdA-induced Bid cleavage was inhibited by calpain and cathepsin inhibitors, but not by caspase inhibitors, suggesting that Bid plays a key role in the caspase-independent apoptosis mechanism [9,47,66,85,86].

Tcds induce apoptosis also, causing alteration in mitochondrial functionality and a low/moderate increase in reactive oxygen species (ROS) by the ROS/JNK/caspase-3 axis [102,103].

However, it has also been demonstrated that Tcds activated pathways of cell death mediated by calpains, likely due to an early increase in intracellular Ca^2+^, with the subsequent involvement of caspases and cathepsins [47,86].

TcdB at high concentrations induces cytotoxicity, mainly by necrosis, characterized by early alteration in plasma membrane permeability, ATP depletion, cellular leakage, and condensation of chromatin without activation of caspases [8,9,10,62,63,64,65,104,105]. TcdB induces necrosis when there is a strong ROS production, likely by lipid peroxidation, oxidation of proteins, damage to DNA, and/or dysfunction of mitochondria [8,9,103]. Remarkably, a TcdB mutant lacking endosomal pore formation did not cause necrotic cell death even at high nanomolar concentrations, suggesting that pore formation is needed for glucosylation-independent necrosis induced by TcdB [8,9,61,93,106]. In contrast, at all concentrations examined, TcdA does not lead to ROS production or induce a low level of ROS, causing cell death mainly by apoptosis [9,63,103].

TcdB also induces cytotoxicity by pyknotic cell death, characterized by ballooning of the nuclear envelope and condensation of chromatin, both glucosyltransferase domain-dependent and -independent, and ROS-independent [8,64,65,104].

Furthermore, TcdB induces a form of cell death called pyroptosis, characterized by a strong inflammation and pore formation mediated by gasdermin-D, which is the substrate of proinflammatory caspases implicated in inflammasome generation [8,9,107,108]. Gasdermin-D, in resting conditions, presents an N-terminal domain joined to its C-terminal domain that behaves as an autoinhibitory factor [109]. The proinflammatory caspases cut the bond connecting the two domains in order that gasdermin-D can bind to the plasma membrane forming pores, which causes strong alteration in ion permeability with cell osmotic swelling and then pyroptosis [105,107,109,110,111].

In this context, the initial key event of assembly and activation of the NLRP3 inflammasome seems to be a rapid Ca^2+^ mobilization, due to Ca^2+^ release from both the endoplasmic reticulum and damaged mitochondria. This release plays a critical role in the apoptotic–pyroptosis cascade, which can be mediated by caspase activation triggered by inflammasome activation [112] and by the Ca^2+^-mediated calpain activation and subsequent activation of caspases and cathepsins [47,86,100].

While the activation of calpains and their primary involvement in the apoptotic cascade induced by TcdB in EGCs has been well documented in vitro [47], there are presently no studies aimed at understanding how intracellular Ca^2+^ flux is activated and/or whether there is mobilization of intracellular Ca^2+^ stores, events necessary for calpain activation.

However, as discussed in the following paragraphs, several studies seem to provide support for a primary role of Ca^2+^ in cell death induced by TcdA and TcdB, suggesting possible mechanisms of the increase in intracellular Ca^2+^ induced by Tcds.

## 4. Alteration in Ca^2+^ Homeostasis by TcdA and TcdB

Most bacterial toxins, either by binding membrane receptors or causing pores on the plasma membrane/membrane of intracellular organelles such as lysosomes and endosomes, can alter cytoplasmic Ca^2+^ homeostasis [113,114,115,116]. Intracellular Ca^2+^ can increase due to an influx from the extracellular environment, release from intracellular stores such as the endoplasmic reticulum and lysosomes, or by both mechanisms [113,114,115]. The cell tries to antagonize the increase in cytosolic Ca^2+^ induced by bacterial toxins by means of Ca^2+^ sequestration mechanisms in the endoplasmic reticulum and mitochondria and by extrusion via selective Ca^2+^ pumps [113,114,115].

At the same time, the cells try to repair the damaged plasma membrane by endocytosis or shear and removal of the damaged portion of the membrane (shedding of microvesicles) [113,114,115,116,117,118,119,120]. In cases of inability to return to homeostasis, the cell can undergo death by apoptosis or necrosis [113,114,115,116].

As previously stated, Tcds induce rapid cell cycle arrest, and depending on the Tcd concentration, Tcd receptors, and cell type, can cause death by apoptosis or necrosis [8,9,10,21,46,59,63]. While alterations in the signal pathways that regulate the cell cycle and cell death by apoptosis or necrosis induced by TcdA and TcdB have been repeatedly and thoroughly studied [8,9,10,11,46,47], the possible role of Ca^2+^ in cell death and more generally its effects on the mitochondria, lysosomes, and Ca^2+^-sensitive molecules have scarcely been investigated. In contrast, several studies are available on the family of PFTs [113,114,115,116], to which Tcds do not belong.

Recent studies seem to suggest an important role of Ca^2+^ in the toxic activity of Tcds. In fact, it has been demonstrated that when TcdB binds to membrane FZD receptors, which affect cytoplasmic Ca^2+^ levels, inhibition of calpains strongly inhibits TcdB-induced apoptosis [47,51,55,56,58]. Furthermore, both Tcds are able to generate a pore in the endocytic vacuole in which they are internalized [8,9,10,11,12,51,93,121,122,123], highlighting their ability to generate pores in membranes, an important mechanism for an alteration in Ca^2+^ homeostasis. Additionally, a possible key role of Ca^2+^ in the activity of Tcds is suggested by several previous observations and studies that in the light of current knowledge can now be better understood and contextualized.

One of the first demonstrations that Ca^2+^ may be involved in the toxic activity of TcdA and TcdB comes from the studies of Giugliano et al. (1984) through the use of Ca^2+^ antagonists. In fact, *C. difficile* filtrates, which cause both the morphological changes and cytotoxicity in Vero cells (monkey kidney), are inhibited when lanthanum chloride (LaCl_3,_ a Ca^2+^ channel blocker) or chlorpromazine hydrochloride (CRP; inhibitor of calmodulin-dependent stimulation of cyclic nucleotide phosphodiesterase) are added to the filtrates. This inhibition is dose-dependent, and when LaCl_3_ and CRP are added together, the protective effect is greater [124]. LaCl_3_ acts as a Ca^2+^ antagonist, while CRP, among other properties, also binds calmodulin. Therefore, these results suggest that Ca^2+^-dependent events are fundamental for the activity of Tcds and that they appear to depend from Ca^2+^ activation of calmodulin [124].

Subsequently, in a more organic study, it was demonstrated in MRC-5 human lung fibroblast cells that the Ca^2+^ channel blockers verapamil and LaCl_3_ protect against the cytopathic effect of TcdB when added before, concomitantly, and up to 30 min after TcdB [125]. Furthermore, two calmodulin antagonists, trifluoperazine and amitriptyline antagonize very effectively the cytopathic effect of TcdB, implicating a role for calmodulin in the toxic activity of TcdB. This inhibitory activity is not due to the possible involvement of calmodulin in endosome–lysosome fusion following endocytosis of TcdB [125]. Thus, Ca^2+^ and calmodulin activation are required for the toxic activity and internalization process of TcdB. In fact, the functionality of Ca^2+^ channels is absolutely necessary for TcdB to exert its toxicity [125].

However, TcdA appears to be different from TcdB regarding the role of Ca^2+^ in its toxic activity. In fact, in intestinal epithelial cells, IEC-6, the early alterations in cytoskeleton surface blebbing and nuclear fragmentation (BERN) are not inhibited by Ca^2+^ channel blockers, inhibitor of Ca^2+^-activated catabolic enzyme, calmodulin antagonists, or Ca^2+^ ionophores [126]. Thus, TcdA-induced cytoskeletal changes are not due to alteration in Ca^2+^ homeostasis. A further interesting phenomenon induced by TcdA is the strong formation of surface blebbing in about 50% of the cells [126]. These results, which show that Ca^2+^ plays no role in TcdA activity on IEC-6 [126], are contrary to those that demonstrated that Ca^2+^ plays a role inTcdA activity on human granulocytes [127,128,129].

However, it is clear that the involvement of Ca^2+^ by TcdA and TcdB in cytotoxic and cytopathic activity can vary according not only to cell type but also to Tcd dose. Indeed, TcdA in human granulocytes induces a rapid and transient increase in cytosolic Ca^2+^ that begins within 1–3 s, reaches a peak at 30 s, and returns to baseline values after about 2 min. This transient increase is not due to the Ca^2+^ influx, but to the mobilization of intracellular Ca^2+^ from intracellular Ca^2+^ reserves, partly mediated by guanine nucleotide-binding protein G(i) [128,129].

In primary cultures of astroglia, TcdB induces morphological changes which require extracellular Ca^2+^ but only for TcdB internalization. However, it is difficult to exclude that during the internalization process, an influx of Ca^2+^ or its exit from intracellular reserves does not occur. That Ca^2+^ may, however, contribute to the toxic activity of TcdB is suggested by the demonstration that the direct introduction of Ca^2+^ ions into the cell via the Ca^2+^ ionophore, A23187, enhances the morphological changes induced by TcdB. A further interesting aspect of these studies is represented by the cells that survive TcdB, which increase the cytoplasmic ramifications much more than the controls after 26 days of culture, and further by the appearance of flat epithelioid astrocytes with showy bundles of filaments suggesting the transition to the state of senescence [130].

The variability in the role of Ca^2+^ in the toxic activity of TcdA is confirmed in a model in which TcdA is given to Chinese hamster ovary (CHO) cells and human colonic tumors cells, T84 [131,132]. In this model, the cytotoxic activity of TcdA is not inhibited by Ca^2+^ channel blockers or reducing agents, leading to the conclusion that the cytotoxicity of TcdA is not a Ca^2+^-dependent activity [131]. This could be because TcdA has several receptors, some expressed only on certain cell types; therefore, the role of Ca^2+^ could depend on the type of receptor involved and the cell type. Thus, some receptors once bound by TcdA have the ability to internalize TcdA and induce toxicity by altering Ca^2+^ homeostasis, while others internalize TcdA without altering Ca^2+^ homeostasis [131].

Another study demonstrated that the cytotoxic and cytopathic activity of TcdB in human fibroblast cells, NIH-3T3, is due to a sustained increase in intracellular Ca^2+^ due to influx of extracellular Ca^2+^ [133]. The cell response to the Ca^2+^ increase follows a bimodal pattern that characterizes two cell populations [133]: (1) 25% of cells characterized by a rapid increase in cytosolic Ca^2+^ concentration (2.5 s) and fast Ca^2+^ influx followed by a sustained steady state; and (2) 75% of cells characterized by a slow increase in cytosolic Ca^2+^ concentration, always followed by a steady state. In fast Ca^2+^ influx, Ca^2+^ initially localizes in discrete regions close to the membrane, where it reaches higher concentrations, then there is a high and homogeneous diffusion of Ca^2+^ in the cytosol [133]. Even in cells in which TcdB causes a slow rise in Ca^2+^, the latter diffuses into the cytosol [133]. While the increase in Ca^2+^ is exclusively due to its extracellular influx, subsequently the Ca^2+^ released from intracellular stores contributes to the increase in Ca^2+^ intracellular concentration [133]. Further confirmation of the need of Ca^2+^ for the toxic activity of TcdB occurs with the sequestration of intracellular Ca^2+^ by 1,2-bis(o-aminophenoxy)ethane-N,N,N′,N′-tetraacetic acid (BAPTA) and Ca^2+^ clamping, which abolishes the cytotoxicity [133]. The same phenomenon is induced by TcdB in other cell types, such as rat pancreatic adenocarcinoma cells, AR42J, and primary cultures of gastric smooth muscle [133].

However, there are other important considerations concerning this phenomenon: (1) Ca^2+^-independent mechanisms, such as cathepsin activation, may contribute to TcdB toxicity [47]; (2) sustained levels of intracellular Ca^2+^ can activate other Ca^2+^-dependent protease such as calpains, which can contribute to cytotoxicity [47] and to modify proteins involved in the regulation of the cytoskeleton and the cell cycle; (3) there is also an involvement of mitochondria, which are profoundly affected by variations in Ca^2+^ homeostasis, especially if induced from the outside [133].

In human monocytes, TcdA induces an upregulation in the secretion of interleukin-8 (IL-8), one of the cytokines responsible for the inflammatory response that accompanies CDI. In particular, TcdA induces IL-8 upregulation by an influx of Ca^2+^ and the mobilization of intracellular Ca^2+^, with the activation of calmodulin. In fact, the association of ethylenediaminetetraacetic acid plus BAPTA completely abrogates TcdA-induced IL-8 secretion, while calpain inhibition abrogates it by about 80% [134]. Furthermore, the fundamental mechanism activated by TcdA involves the activation of nuclear factor kappa-light-chain-enhancer of activated B cells (NF-κB) and its translocation to the nucleus in the p50/p65 heterodimeric isoform [134,135,136].

Extracellular Ca^2+^ is required for binding of the cell wall binding domain of TcdA to plasma membrane of CHO cells [137]. In addition, the cytotoxicity of TcdA at low concentrations is also dependent from an increase in intracellular and extracellular Ca^2+^. Conversely, TcdA at high concentrations has a toxicity independent from Ca^2+^. Thus, Ca^2+^ plays a role in TcdA binding to the target cell and renders this more sensitive to TcdA cytotoxicity [137].

Most PFTs, in addition to creating pores in cell membranes, creates them in the mitochondrial membranes [113,114,115]. Direct targeting of mitochondria allows bacteria to bypass key steps controlling cell death and thus directly target one of the organelles critical for execution of apoptosis [95,96,113,114,115]. It is worth noting that both Tcds are capable of perturbing mitochondrial function [8,9,10,11,12,48,138]. TcdB in human epithelial cells, Hep-2, causes an early hyperpolarization of mitochondria, followed by a limited increase in cytosolic Ca^2+^ that precedes the final stage of apoptosis [91]. Actually, already after 30–35 min from internalization, TcdB induces an increase in cytosolic Ca^2+^ caused by an influx, which is indispensable for altering mitochondrial functionality and the subsequent release of cytochrome C [91].

TcdB added in vitro to isolated mitochondria causes their swell and releases cytochrome C only in the presence of Ca^2+^. Furthermore, it is possible that the Ca^2+^ increase induced by TcdB blocks the mitochondrial ATP-sensitive potassium (K+)-channel, thus contributing to apoptosis [91].

Ca^2+^ also has a very important role in the necrotic response of Hela cells to relatively high concentrations of TcdB (≥100 pM) due to aberrant production of ROS following the assembly of NADPH oxidase (NOX). These high levels of ROS cause mitochondrial damage, lipid peroxidation, and protein oxidation. This phenomenon depends on the activity of L-type Ca^2+^ channels, because most of their inhibitors block the cytotoxic activity of TcdB [9,64].

The Ca^2+^ response mechanism to TcdB is characterized by two phases, with an initial peak of Ca^2+^ increase in 10–15 min, which returns to baseline levels after a further 20 min (total about 30 min). This signal is quite small, but it is followed by a second increase at 40–45 min that then persists over time. The increase in intracellular Ca^2+^ is due to the influx of extracellular Ca^2+^ [82,139]. Furthermore, after 15 min from stimulation with TcdB, there is an increase in activity of protein kinase C (PKC) [82]. Thus, TcdB induces an influx of extracellular Ca^2+^ through L-type Ca^2+^ channels that leads to the rapid activation of PKC with consequent aberrant activation of NOX, which causes the production of a high level of ROS and necrosis as a final event [64,82].

In another experimental model, at 12 h, TcdB induced a strong increase in intracellular Ca^2+^, also dose-dependent. This increase is due to the upregulation of NOX4, which is mainly responsible for alterations in Ca^2+^ homeostasis and subsequent endoplasmic reticulum stress. Thus, NOX4 activated by TcdB acting on Ca^2+^ homeostasis is mainly responsible for cytotoxicity and apoptosis [140].

Other studies highlight the complexity of the interaction between TcdB and Ca^2+^. In fact, in the synaptosomes, where the alpha toxin creates pores allowing the influx of extracellular Ca^2+^ with the consequent stimulation of phospholipase D, TcdB inhibits the activation of phospholipase D by blocking Rho-GTPase. These results indicate that the Ca^2+^ influx induced directly by TcdB can be subsequently stopped by TcdB itself through inhibition of Rho-GTPase, which also causes inhibition of phospholipase D, thus preventing the formation of choline and phosphatidic acid [61,141]. Then, when TcdB has completed the inhibition of Rho-GTPases, the process of inhibition of the increase in intracellular Ca^2+^ due to extracellular influx and/or mobilization of intracellular Ca^2+^ stores starts. However, the strongest inhibitory effect is on Ca^2+^ influx [142]. Of course, it is possible that the role of Rho-GTPases in the regulation of Ca^2+^ also depends on the cell type. A relationship between TcdB-induced increase in Ca^2^ signaling and the phospholipase family is suggested in a macrophage cell model, RAW 264.7, where the phenomenon is independent of Rho-GTPase [61,143].

The cytotoxicity of Tcds depends on the cell type because they express different receptors; however, Tcds have the ability to cause cell death in all, further underlining a possible common mechanism of action that could be mediated by Ca^2+^ alterations.

It has been shown that alterations in Ca^2+^ homeostasis occur before significant Rho-GTPase inhibition. However, it is possible that after the inhibition of Rho-GTPases, a further alteration in Ca^2+^ homeostasis occurs, contributing to a stronger activation of Ca^2+^-dependent molecules, including calpains. This makes the cell death process, which is induced by different molecular mechanisms, irreversible and explains the frequent observation that the inhibition of a single death pathway, for example, that mediated by caspases, does not completely prevent cell death.

Loureiro et al. [144] demonstrated that *C. difficile* and its Tcds induce activation of pannexin 1 channels, which favor cell death via release of ATP, activation of purinergic receptors and increased intracellular Ca^2+^ flow, causing the activation of initiator and effector caspases and leading to apoptosis [144].

Kim et al. demonstrated that TcdA induced colonocytes apoptosis by the release of prostaglandin E2 which by stimulating EP1 receptors induce PLC activation, Ca^2+^ mobilization, and PKC activation [145].

Loureiro et al. [146] analyzed the modulation of response of host and *C. difficile* by nutrients, including Ca^2+^, that play a key role in sporulation and germination of *C. difficile* in the gut [146], reporting that there was only one recent paper describing a role of Ca^2+^ in vitro [82] and that no reports on Ca^2+^ uptake or serum Ca^2+^ levels in CDI in patients or murine models.

Overall, the studies on the relationship between Tcds and Ca^2+^ suggest that in Tcd activity, there is an early phase of about 30–40 min in which Ca^2+^ homeostasis is altered, with effects that depend on Tcd concentration, cell type and Tcd receptors involved. The increase in intracellular Ca^2+^ concentration, both distributed throughout the cell and with localized effects on substructures such as vacuoles and vesicular compartments, may be responsible for events crucial for cell survival, such as alteration in mitochondrial polarization, activation of calpains, and alterations in endosome stability, by which the cathepsins are kept under control. There is also the ability of calpains to activate caspases. Then, a series of Ca^2+^-dependent events fundamental for cell survival takes place, before the occurrence of other events, after about 40 min, resulting from the inhibition by Tcds of Rho-GTPase. These latter are involved in cell cycle arrest and the activation of signaling pathways, which add to the effects already caused by the alteration in Ca^2+^ homeostasis.

All the data above suggest that in their interaction with numerous types of tissue, nervous cells and immune cells, Tcds utilize two important pathways overlapping each other over time, represented by the alteration in Ca^2+^ homeostasis and the inhibition of the Rho-GTPase family. This in order to be more capable of counteracting the various types of survival pathways in each cell type, including immune cells, the latter with different Tcd susceptibility due to their functional state of activation.

It is important to remember that Tcds have a further strategy for neutralizing cells that have resisted toxic activity, i.e., inducing a state of senescence [80], in which the possible role of Ca^2+^ has yet to be investigated.

Unfortunately, there have been no in vivo studies on this issue.

## 5. Comparison between PFT and Tcd Properties

### 5.1. Formation of Pore/Ca^2+^ Channels on the Plasma Membrane

Large clostridial toxins (LCTs) are all PFTs, i.e., toxins that cause pores in the plasma membrane through which occurs an influx of Ca^2+^ that alters its cellular homeostasis, with important consequences on cell fate [48,113,114,115,116,138]. Studies have also led to the discovery of fundamental mechanisms of membrane pore repair caused by LCTs for the recovery of intracellular Ca^2+^ homeostasis [114,138]. Although Tcds do not belong to the PFTs, several studies reported above have suggested that the alteration in Ca^2+^ homeostasis could have a central role in activating cell death signaling in addition to that derived from Rho-GTPase inhibition [114,116].

In this regard, comparison between the mechanism of cell death activated by PFTs and that of Tcds is extremely interesting, because of the many similarities that further encourage us to look at Tcds as molecules able to alter intracellular Ca^2+^ homeostasis (Table 1).

The comparison between PFTs and Tcds considers that PFTs form a stable pore in the membrane that becomes mainly responsible for the influx of Ca^2+^ causing profound alterations in intracytoplasmic Ca^2+^ homeostasis, while the pathogenic activity of Tcds has been characterized only as the activation of Rho-GTPase-dependent and Rho-GTPase-independent mechanisms after internalization of the Tcd.

However, other than direct demonstrations of an influx of Ca^2+^ in cells exposed to Tcd, many Ca^2+^-dependent effects have been described, which indirectly demonstrate the presence of an important alteration in Ca^2+^ homeostasis as activating factor (Figure 1), very similar to that described for PFTs.

Therefore, the key point in the comparison between Tcds and PFTs is not the transient pore that Tcds create with respect to a “stable” pore caused by PFTs, but the Ca^2+^-dependent effects of Ca^2+^ alone and on Ca^2+^-dependent molecules. This reinforces the concept of the role of Ca^2+^ in the pathogenesis of Tcds. In fact, various molecules activated by PFTs with a clear influence of Ca^2+^ have been compared with those activated by Tcds, also offering speculative insights on the repair mechanisms that attempt to antagonize the alteration in Ca^2+^ homeostasis.

The constituting monomers of PFTs, after binding to their receptors, assemble in multimers, forming a functional transmembrane pore [113,114,115,116]. The formation of pores on plasma membrane alters the mechanisms that control the major differences between the interior of the cell and the extracellular environment where high extracellular Ca^2+^ concentration causes an influx, whereas at the same time high intracellular K+ concentration prompts its outflow. Thus, in such conditions, Ca^2+^ enters the cell by passive flux because PFT has caused plasma membrane pores [113,114,115,116]. The effects on the cell will then depend on the extent of the Ca^2+^ influx and thus on the increase in its intracellular concentration, duration of the influx, type of cell involved, Ca^2+^ extrusion mechanisms activated and their effectiveness, and the efficacy of pore repair induced by PFTs [113,114,115,116].

Concerning Tcds, there are no data demonstrating a direct ability to form pores on cell plasma membranes. However, indirect data suggest that Tcds interact with plasma membranes by altering their symmetry and therefore permeability to ions such as Ca^2+^ (Figure 1). In fact, TcdA creates pores on artificial lipid membranes [93]. Furthermore, both Tcds, once inside the endocytic vacuole, have a domain capable of creating a pore for the exit of the catalytic domain, even though this property is due to a conformational change in the Tcds caused by the pH lowering of the endocytic vacuole following the activity of vacuolar ATPases (Figure 1) [8,9,10,11]. In addition, it is possible that after the binding of Tcds to their receptors, the internalization process in the endocytic vacuoles, whose membrane is an invagination of the plasma membrane, creates transient pores through which a rapid and transient influx of extracellular Ca^2+^ can be formed (Figure 1) [8,9,10,11,51,93,121,122,123,147]. Therefore, even if in a more indirect and transient way, Tcds can also induce an influx of Ca^2+^ by acting on the cell membrane (Figure 1).

In the case of PFTs, the influx of Ca^2+^ can be massive due to the strong concentration gradient between the extracellular environment and the cytosol [113,114,115,116]. In this case, Ca^2+^ entry has monophasic kinetics, but if the cell is able to pour out the excess Ca^2+^, the kinetics become biphasic [113,114,115,116]. In various models utilizing Tcds, the increase in kinetics of Ca^2+^ entry in the cytosol is rapid and often followed by a drop, characterizing it mainly as biphasic [133]. Thus, it is expected that pores created by Tcd endocytosis can be rapidly repaired, as well as plasma membrane alterations, if they are followed by an effective process of membrane repair.

### 5.2. Activation of Ca^2+^ Channels on the Plasma Membrane

This mechanism contributes to the direct influx of Ca^2+^ from the outside, but unlike the pore, it operates selectively. Both PFTs and Tcds activate specific Ca^2+^ channels on the membrane, as convincingly demonstrated with Ca^2+^ channel inhibitors [82,142,148,149].

### 5.3. Activation of Ca^2+^ Channels in the Endoplasmic Reticulum

The activation by PFTs of these channels, able to selective release Ca^2+^, is based on the stimulation of the inositol triphosphate receptor (IP3R) by inositol triphosphate (IP3). This latter is generated through direct or indirect stimulation of G proteins at the plasma membrane level with the subsequent activation of phospholipase A2, favoring the generation of IP3. This in turn stimulates its receptor on the endoplasmic reticulum and final release of Ca^2+^ [150,151,152,153,154]. 

There are no data available on the possible interaction of Tcds with G proteins; however, TcdA and TcdB are capable of activating phospholipase A2 and therefore of generating IP3 [155,156].

### 5.4. Activation of Ca^2+^ Channels in Lysosomes

It has been suggested that PFTs might induce Ca^2+^ release from lysosomes through two-pore channels following plasma membrane stimulation of CD38. This activates the pathway involving nicotinic acid adenine dinucleotide phosphate (NAADP), responsible for Ca^2+^ release [114]. Damage to lysosomes with consequent non-selective Ca^2+^ release can contribute to PFT activity [114].

While there are no data on the activity of Tcds on the CD38-NAADP lysosomal pathway, Tcds are able to act on lysosomes, altering their membrane functionality with release of cathepsins [47,86,123,157], thus implicating possible alterations in lysosomal Ca^2+^ homeostasis.

### 5.5. Activation of Ca^2+^-Dependent Cytoplasmic Proteins

Ca^2+^-dependent cytoplasmic proteins, if activated, are very sensitive sensors of alterations in Ca^2+^ homeostasis, and thus of the presence of pores, alterations in the Ca^2+^ channels of the endoplasmic reticulum, and/or alterations in the Ca^2+^ channels in lysosomes. In fact, PFTs activate calpains, calmodulin, calcineurin, PKC, and phospholipase [114,158,159,160,161,162].

As further confirmation of the role of Ca^2+^ in the activity of Tcds, TcdB and to a lesser extent TcdA are able to activate these Ca^2+^-dependent cytoplasmic proteins. For instance, calpains are activated by TcdB in EGCs [47]. TcdB is able to activate calmodulin [163], while TcdA and TcdB activate phospholipase A2 [155,156]. Both TcdB and TcdA activate PKC alpha/beta [164,165].

### 5.6. Alteration in Ca^2+^ Homeostasis and Cell Death

The profound alterations in Ca^2+^ homeostasis caused by PFTs activate cell death pathways [113,114,115,116] that are very similar to those activated in particular by TcdB. However, it is clear that the intracellular Ca^2+^ levels reached, the capacity of Ca^2+^ extrusion mechanisms, and the cell type and receptor involved are the elements that decide the prevailing type of cell death.

#### 5.6.1. Necrosis

The main events characterizing cell death by necrosis induced by PFTs and mediated by Ca^2+^ are the activation of calpains and calmodulins with consequent activation of caspases. Cathepsins released from lysosomes after functional alteration in their membrane also participate to the necrosis process, and to these events should be added the effects of release of ROS [101,113,166]. The final consequence is the lysis of the cell that has undergone the process of necrosis [114,116,167].

Even Tcds in certain situations can induce cell death by necrosis [8,9,10,62,63,64,65,93,106]. In fact, TcdB activates calpains and cathepsins even at low doses [47,86]. Furthermore, TcdB activates calmodulin [163] and both Tcds induce ROS, although to a different extent [8,9,10,102,103].

#### 5.6.2. Apoptosis

PFT-induced apoptosis is characterized by a mitochondrial pathway with release of AIF and/or cytochrome C from the mitochondria and by a caspase-dependent pathway mainly triggered by Ca^2+^-activated calpains and calmodulins [113,114,115,116].

Tcds are capable of inducing apoptosis [8,9,10,11,46,47,60] through a mitochondrial pathway and cytochrome C release, which activates caspase-9 and the executioner caspases (caspase-3, caspase-6 and caspase-7) [46,47,66,85,86,87,93,94,102,103]. Tcds are also able to induce apoptosis by activated calpains, with effector expression in caspases [47,86].

#### 5.6.3. Pyroptosis

The signal that starts from the mitochondria is mediated by NLRP3 in PFTs and likely also in Tcds and depends on Ca^2+^ mobilization. In fact, it has been reported that Ca^2+^ mobilization activates NLRP3 inflammasome [112]. The latter requires Ca^2+^ signaling/mobilization, which can be promoted by several cellular changes induced by pathogens or their toxins, such as K+ efflux and phagolysosome alteration, thus being responsible for ATP-mediated mitochondrial Ca^2+^ damage. In fact, it has been demonstrated that Ca^2+^ signal inhibitors block ATP-mediated mitochondrial Ca^2+^ damage by evaluating mitochondrial ROS production, loss of mitochondrial membrane potential, and release of mitochondrial DNA into the cytosol. Since these processes are needed for NLRP3 inflammasome activation, the critical role of Ca^2+^ signaling/mobilization in NLRP3 activation is to mediate mitochondrial damage [112]. However, other than induction of pyroptosis by direct activation by caspase-1 of executioner caspases [8,9,105,107,109,110,111,112], induction of cell death (pyroptosis/apoptosis) can also be hypothesized to occur through a mitochondrial pathway by alteration in Ca^2+^ homeostasis and calpain activation, effects induced by both PFTs and Tcds.

#### 5.6.4. Comparison between Aerolysin and Tcds

An organic view of cell death mechanisms induced by the most studied PFTs is provided by aerolysin, an orthologue of the alpha toxin of *C. septicum* whose activated signal pathways are very analogous to those induced by TcdB and TcdA. The first decisive event is the alteration in Ca^2+^ homeostasis, with its increase, rapidly followed by the activation of calpains, which act on the stability of the lysosomal membrane, favoring the release of cathepsins acting on various substrates [113,114,116,167,176]. Furthermore, calpains activate caspases and act on proteins that form Na^+^/Ca^2+^ exchange pumps, resulting in further Ca^2+^ influx. This causes an alteration in MOMP that leads to an increase in ROS and a decreased production of ATP. Finally, DNA damage occurs and activates PARP, which in turn contributes to ATP depletion and thus to cell death [113,114,116,167,176]. Of course, the dose of aerolysin and the type of cell involved dictate whether the final characteristics of cell death are apoptotic or necrotic [113,114,116,167,176].

This picture of cell death is very similar to that caused by Tcds, especially by TcdB, as shown from studies demonstrating that the initial rapid event is disruption of Ca^2+^ homeostasis, followed by calpain activation and release of cathepsin from lysosomes [47,86,123,157]. Subsequent caspase activation contributes to signaling pathways leading to PARP activation and DNA damage with ATP depletion [46,47,86,91,102]. Simultaneously, the various functional alterations in the mitochondria lead to a strong increase in ROS, with cell death by apoptosis or necrosis depending on the initial Tcd concentration and the cell type [8,9,10,102,103].

## 6. Repair Mechanisms of the Influx of Ca^2+^

Under increased cytosolic Ca^2+^, the cell responds with different types of mechanisms to lower the cytosolic ion concentration and to avoid cell death [114,115,116,138]: (1) Ca^2+^ extrusion by plasma membrane pumps, Ca^2+^-ATPase; (2) sequestration of Ca^2+^ within the endoplasmic reticulum reserves by sarco/endoplasmic reticulum Ca^2+^-ATPase; (3) and sequestration of Ca^2+^ in mitochondria by the mitochondrial Ca^2+^ uniporter. If the sequestration becomes excessive, it can lead to intoxication of the mitochondrion itself.

These events aimed at reducing the cytosolic Ca^2+^ load may be effective whether the influx of Ca^2+^ is slowed or stopped, acting on the pores to eliminate them [114,115,116,138].

There are three main mechanisms of pore clearance, as follows.

Ectocytosis: this is a process of elimination towards the outside of the membrane areas that contain the pores by formation of microvesicles, and it takes place by two modalities [114,115,116]. The first is based on the activity of the annexins, which—activated by the high concentration of cytosolic Ca^2+^—migrate to the plasma membrane where the pores have formed [117]. Here, annexins establish a firm bond with Ca^2+^, contributing to the formation of a strong interaction with the plasma membrane. The latter forms membrane folds that are subsequently expelled, detaching from the membrane, which then rapidly reseals the externalized microarea called “microvesicles (MVs)” [117,118,119]. The second modality is realized through the activity of the endosomal sorting complex required for transport (ESCRT) [177,178].Multivesicular bodies: this is one of the membrane repair mechanisms that provokes a profound alteration in Ca^2+^ homeostasis, with an increase in the cytoplasmic concentration creating MVs that have two main purposes [118,119]. The first is to seal the parts of the membrane in which the pores have formed, mainly responsible for the influx of Ca^2+^ by ectocytosis; the second is to surround the cell with MVs derived from the membrane containing the receptors for the PFTs. Therefore, the PFTs are bound before they can reach the cell membrane. Despite the fact that Tcds have been convincingly shown to alter Ca^2+^ homeostasis, there are no direct studies on MV formation in response to Tcds. However, an interesting study on platelets demonstrated that a pathway that regulates the formation of MVs is blocked by TcdB [179]. This could explain why Tcds, despite the induction of Ca^2+^ increase, at the same time prevent the cell to form MVs as a protective response. In particular, the induction of a Ca^2+^ influx in platelets leads to the formation of MVs through the Rac1 and p21-activated kinase 1/2 (PAK1/2) activation pathway with the involvement of the activation of calpains, which in turn activate caspases [179]. By inhibiting Rac1, TcdB blocks the formation of MVs. Therefore, if TcdB induces Ca^2+^ influx, the cell’s response should be to generate MVs, but this cannot happen because TcdB inhibits Rac1 [179].This aspect is relevant for Tcd pathogenesis, because if the MV response also has a protective role [179], TcdB could have chosen a very sophisticated strategy to prevent the cell from protecting itself from TcdB by MVs.Endocytosis: this is a process by which the portion of the membrane that contains the pore is internalized and sent to the lysosome, where it is degraded [114,115,116]. Both endocytosis and ectocytosis require the participation of the regulatory molecules RAB-5 and RAB-11 [114,115,116].

It is also evident that in the cases of PFTs and Tcds, there is always in a given population of cells a percentage that does not undergo any form of cell death, suggesting that in some of them are active effective mechanisms of intracellular Ca^2+^ sequestration and/or the pore repair process.

## 7. Destruction of Intercellular Junctions (IJs)

IJs are one of the primary targets of PFTs, and the main consequence of their destruction is bacterial transmigration across the tissue barrier and the deepening of the infection [113,114,115,180].

Ca^2+^ plays a central role in this process of IJ destruction and on the consequent redistribution of ZO-1 [113,114,115,180]. Under resting conditions, intracellular pro-ADAM10 (a protease transmembrane, the main substrates of which are adhesion molecules, E- and VE-cadherin, expressed at the epithelial and endothelial junctions, respectively) is bound to calmodulin, blocking cleavage and activation by furin [114,180,181]. However, when there is an increase in cytosolic Ca^2+^, there is consequent activation of calmodulin and the release of pro-ADAM10, which can be activated by furin. At this point, activated pro-ADAM10 is transferred to the cytoplasmic membrane, where it causes shedding of cadherin and therefore IJ destruction [114,180,181]. Among the substrates of pro-ADAM10 are receptor kinases and matrix proteins [114,180,181]. Another mechanism by which increased intracellular Ca^2+^ can alter the permeability of the barrier is mediated by the activation of the myosin light chain kinase pathway [114,180].

IJs are an important target of Tcds because their destruction favors the deepening of the infection in the submucosal colonic layers [8,9,10,11,60,172,173,174,175]. In particular, TcdB activates calmodulin and causes shedding of cadherin [168] and redistribution of ZO-1 [9,60,164,170,171,172,173]; TcdA impairs IJs mainly causing redistribution of ZO-1 [9,60,169,172,173,174].

## 8. Conclusions

CDIs represent an important and serious widespread health problem, with an increasing incidence in both Western and Eastern communities. This increase is still under investigation, but some data suggest that it might be associated with the appearance of hypervirulent epidemic isolates of ribotype 027 [182] or with advancing methods of species identification and susceptibility testing [183].

The most important event in CDI is represented by cell death caused by TcdA and TcdB, which is then followed by inflammation with immediate and long-term consequences.

Important results have been obtained in studies concerning the complex activation mechanisms and signaling pathways that lead to various types of cell death, apoptosis, necrosis, pyknosis and pyroptosis. Given that Tcd dose, cell type, Tcd receptors involved and their expression in various cell types could account for the complexity of the various signaling pathways activated, perhaps it is possible to identify in the alteration in intracellular Ca^2+^ homeostasis the early key event for the activation of the various types of cell death.

Of course, if Ca^2+^ is the key player in the activation of cell death, the alterations in its homeostasis in terms of duration, Ca^2+^ concentration and localization in the various cellular microenvironments can in turn explain the great complexity of the activated signaling pathways involved in the various types of cell death. In particular, the inactivation of Rho-GTPases by Tcds may block the mechanisms that could attempt to counteract the increased intracellular Ca^2+^ concentration.

Additionally, it is interesting that TcdA and TcdB mechanisms and effects on intracellular Ca^2+^ homeostasis are quite similar to those elicited by PFTs, which base most of their toxic action on the direct influx of Ca^2+^ through membrane pores to alter intracellular Ca^2+^ homeostasis. Further studies of the key role of Ca^2+^ in the pathogenetic mechanisms used by Tcds in CDI are needed and could be useful to explore new therapeutic strategies, focused on these early pathogenetic mechanisms, in order to prevent or block them, and to stop the subsequent cascade of events leading to this important inflammatory state of the gut.

## Figures and Tables

**Figure 1 biology-12-01117-f001:**
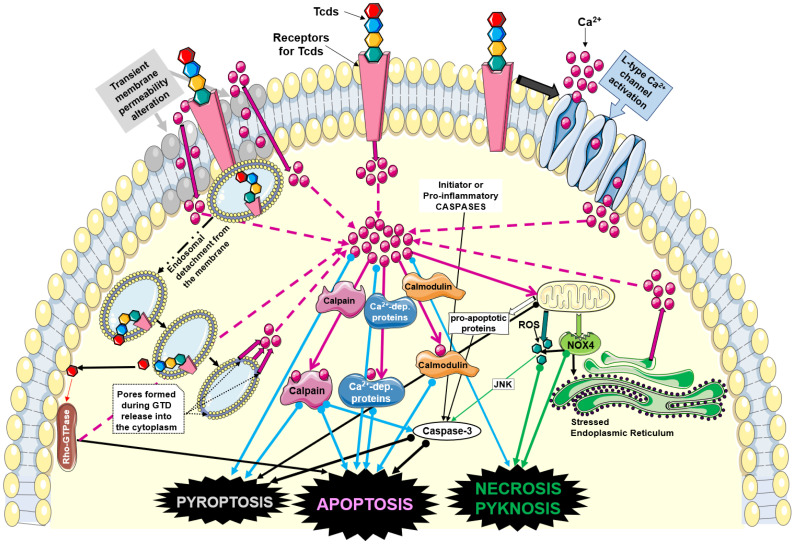
Mechanisms of Ca^2+^ homeostasis alterations induced by Tcds and involved in induction of different types of cell death. Abbreviations: *Clostridioides difficile* (*C. difficile*) toxins (Tcds); calcium (Ca^2+^); NADPH oxidase 4 (NOX4); reactive oxygen species (ROS); N-terminal glucosyltransferase domain of Tcds (GTD). Solid pink arrows indicate activation; solid thick arrow indicate influx or release; dotted pink arrows indicate Ca^2+^ increase; dotted black arrows indicate pore formation. Green, cyan, and thick black arrows with full circle indicate activation of cell death pathways and cell death types involving Ca^2+^. The figure was partly drawn using pictures from Servier Medical Art (https://smart.servier.com/image-set-download, last accessed on 5 June 2023). Servier Medical Art by Servier is licensed under a Creative Commons Attribution 3.0 Unported License (https://creativecommons.org/licenses/by/3.0/, last accessed on 5 June 2023).

**Table 1 biology-12-01117-t001:** Comparison of PFTs and Tcds on mechanisms and consequences of Ca^2+^ homeostasis alterations.

Mechanism	PFTs	Tcds
Formation of Ca^2+^ channels on the plasma membrane	PFTs monomer polymerizes within the plasma membrane, forming a channel that favors the influx of Ca^2+^ based on the extracellular gradient of Ca^2+^ which is greater than the intracellular one Refs. [48,113,114,115,116,138]	Tcds do not form pores in the plasma membrane but can activate the influx of Ca^2+^ by stimulating the receptors as FZDs. In addition, transient changes in the continuity of the plasma membrane during the endocytosis process of Tcds can favor Ca^2+^ influx Refs. [8,9,10,11,51,93,121,122,123,133,147]
Activation of selective Ca^2+^ channels in the plasma membrane	Demonstrated with inhibitors Refs. [48,113,114,115,116,138]	Demonstrated with inhibitors Refs. [82,142,148,149]
Activation of Ca^2+^ channels in the endoplasmic reticulum	PFTs induce Ca^2+^ release into the cytoplasm by involving IP3 and IP3R, membrane G-proteins, phospholipase A2 Refs. [48,113,114,115,116,138,150,151,152,153,154]	There are no data on the interaction between Tcds and G-proteins. However, TcdA is capable of activating phospholipase A2 and thus to generate IP3 Refs. [155,156]
Activation of Ca^2+^ channels in lysosomes	PFTs induce Ca^2+^ release from lysosomes by the formation of two-pore channels and the involvement of CD38 and NAADP Refs. [48,113,114,115,116,138]	Tcds alter lysosomal function as demonstrated by cathepsin release implicating alterations in lysosomal Ca^2+^ homeostasis Refs. [47,86,123,157]
Activation of Ca^2+^-dependent cytoplasmic proteins	PFT-induced Ca^2+^ influx activates calpains, calmodulin, calcineurin, PKC and phospholipase A2 Refs. [48,113,114,115,116,138,158,159,160,161,162]	Alteration in Ca^2+^ homeostasis activates calpains, calmodulin, PKC and phospholipase A2 Refs. [47,125,143,155,156,163,164,165]
Cell Death	PFTs induce: Necrosis Necroptosis Apoptosis Pyroptosis Refs. [48,101,113,114,115,116,138,166,167]	Tcds induce: Necrosis Apoptosis Pyroptosis Refs. [8,9,10,11,12,46,47,61,62,63,73,82,84,86,91,94,102,103,104,105,107,108,110,111,126,127,128,129,130,131,132,133,140,142,144,145]
Destruction of the intercellular junctions	Intercellular junctions are targets of PTFs Refs. [48,113,114,115,116,138]	Intercellular junctions are targets of Tcds Refs. [8,9,10,11,60,164,168,169,170,171,172,173,174,175]

Abbreviations: pore-forming toxins (PFTs); *Clostridioides difficile* (*C. difficile*) toxins (Tcds); calcium (Ca^2+^); frizzled proteins (FZDs); inositol triphosphate (IP3); nicotinic acid adenine dinucleotide phosphate (NAADP); triphosphate receptor (IP3R); protein kinase C (PKC).

## Data Availability

Not applicable.

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
