# Peer review of "Role of the Alteration in Calcium Homeostasis in Cell Death Induced by Clostridioides difficile Toxin A and Toxin B"

_biology, 2023, doi:10.3390/biology12081117_

Round 1

Reviewer 1 Report

In this review article, the authors described and highlights for the first time the possibility that Ca2+ homeostasis alteration is the key event of the mechanism of pathogenicity of C. difficile, and the knowledge of this aspect could be of crucial importance to understand how to antagonize this key event of the pathogenicity of C. difficile.

Comments

This is an interesting review article. This manuscript is well-writing. The reviewer has only some minor concerns as follows:

1. In Table 1, please consider including references for the convenience of the reader. Moreover, in Table 1, “2+” for Ca can be superscripted.

2. In Figure 1, please consider adding the various cell death types/pathways to make this diagram for mechanisms of Ca2+ homeostasis more complete.

3. Some abbreviations appear only once, such as CDT (line 54), EDTA (line 405), and MCU (line 666), please consider not using the abbreviation in this situation.

Reviewer 2 Report

Fettucciari et al review the in vitro effect of C. difficile concentrating on the Ca effect and signaling pathways.

This is well covered, and the findings summarized in a meaningful figure. 

The reviewer understands the complexity of the interaction between the bacterium with the GI tract in association with the flora and the effects of the toxin.

It would be interesting considering the panel of the authors and the expected experience to provide some indications as to why “the incidence is increasing in the community, in both elderly and young subjects “

A few ideas would be welcome in the “conclusion” possibly helping to prevent or minimize infection with C. difficile.

Reviewer 3 Report

This manuscript reviewed the role of calcium homeostasis in cell death induced by clostridium difficile toxin A and B. Authors reviewed the Tcds in general, then focused on the alternation of calcium homeostasis induced by Tcds (TcdA and TcdB). Authors reviewed the calcium homeostasis changes followed by the intoxication of Tcds, however, they prematurely linked the changes in calcium homeostasis to Tcds. Many of the references authors cited are from the 1980s to 1990s and from cell cultures (many not from the intestine cell lines, which are more relevant to Tcds). They should focus on more recent literature. It is known that the cytotoxicity of Tcds highly depends on the cell types. Tcds inhibit the Rho-GTPases, and calcium is sensitive to Rho-GTPases, therefore, it is possible that the changes of calcium homeostasis are the results of the inhibition of Rho and the cascades after that. Authors should review to see if there are any in vivo studies that support their hypothesis. While there are more investigations needed to understand the role of calcium in the pathogenetic mechanisms used by Tcds, it is premature to link this to potential new therapeutic strategies for CDI. Overall, authors just listed most of the findings from the literature without critically directing them, and the literature cited is over 20 years old. They should digest those more carefully, and include more from recent literature. This review provides some more information to the readers but needs some revision to reflect the most updated knowledge on Tcds, and the cause of the change of calcium homeostasis.

Overall English is fine and just need some minor changes. 

Reviewer 4 Report

Many thanks to the opportunity to read the paper. I find it well wrote and very clear. 

Sincerly in my opinion is not easy give some suggestion due to high quality of presentation of paper. 

I can only suggest to add e underline the role of education on Cl difficile in health staff due to reduce transmission and hospital outbreak (see Nurses' Knowledge, Attitudes and Practices on the Management of Clostridioides difficile Infection: A Cross-Sectional Study. Antibiotics (Basel). 2023 Mar 7;12(3):529. doi: 10.3390/antibiotics12030529)

Also minor English revision is needed. 

Congratulations to the authors for the excellent paper

Round 2

Reviewer 3 Report

The authors answered most of the questions. However, as authors pointed out, most studies on the role of calcium in the pathogenicity mechanism of Tcds is concentrated in the 80s and 90s, and most studies were based on in vitro studies. There might be reasons for that. The role of alteration of calcium homeostasis in cell death induced by TcdA and TcdB is only a hypothesis, and still needs to be proved. Authors’ claim that “This would pave the way pharmacological interventions aimed at inhibiting this alteration which, if blocked, could arrest the pathogenicity until specific antibiotic therapy has brought the C. difficile infection under control” is ambitious, and probably need to turn down a bit.

Need some minor improvements. 
